# Dose Optimization of Anxiolytic Compounds Group in *Valeriana jatamansi* Jones and Mechanism Exploration by Integrating Network Pharmacology and Metabolomics Analysis

**DOI:** 10.3390/brainsci12050589

**Published:** 2022-04-30

**Authors:** Chengbowen Zhao, Xiaojia Wei, Jianyou Guo, Yongsheng Ding, Jing Luo, Xue Yang, Jiayuan Li, Guohui Wan, Jiahe Yu, Jinli Shi

**Affiliations:** 1School of Chinese Materia Medica, Beijing University of Chinese Medicine, Beijing 102488, China; lovyeazhao@163.com (C.Z.); weixj471@163.com (X.W.); 18811795287@163.com (Y.D.); ljing314@163.com (J.L.); ruixue226@126.com (X.Y.); ljy15612930585@163.com (J.L.); chenxihui912@126.com (G.W.); yujiaheee@163.com (J.Y.); 2Dongzhimen Hospital, Beijing University of Chinese Medicine, Beijing 100007, China; 3CAS Key Laboratory of Mental Health, Institute of Psychology, Chinese Academy of Sciences, Beijing 100083, China; guojy@psych.ac.cn

**Keywords:** *Valeriana jatamansi* Jones, anxiety disorder, network pharmacology, metabolomics, arachidonic acid metabolism

## Abstract

Anxiety disorder impacts the quality of life of the patients. The 95% ethanol extract of rhizomes and roots of *Valeriana jatamansi* Jones (Zhi zhu xiang, ZZX) has previously been shown to be effective for the treatment of anxiety disorder. In this study, the dose ratio of each component of the anxiolytic compounds group (ACG) in a 95% ethanol extract of ZZX was optimized by a uniform design experiment and mathematical modeling. The anxiolytic effect of ACG was verified by behavioral experiments and biochemical index measurement. Network pharmacology was used to determine potential action targets, as well as predict biological processes and signaling pathways, which were then verified by molecular docking analysis. Metabolomics was then used to screen and analyze metabolites in the rat hippocampus before and after the administration of ZZX-ACG. Finally, the results of metabolomics and network pharmacology were integrated to clarify the anti-anxiety mechanism of the ACG. The optimal dose ratio of ACG in 95% ethanol extract of ZZX was obtained, and our results suggest that ACG may regulate ALB, AKT1, PTGS2, CYP3A4, ESR1, CASP3, CYP2B6, EGFR, SRC, MMP9, IGF1, and MAPK8, as well as the prolactin signaling pathway, estrogen signaling pathway, and arachidonic acid metabolism pathway, thus affecting the brain neurotransmitters and HPA axis hormone levels to play an anxiolytic role, directly or indirectly.

## 1. Introduction

Anxiety disorder is a mental disease characterized by persistent, excessive worry, and anxiety symptoms [1]; with the increase in social stress, the incidence rate of anxiety disorders continues to increase, especially during the COVID-19 pandemic. According to a report published by the U.S. Census Bureau, adults in 2020 were four times more likely to have anxiety disorder compared to those in 2019 [2]. People of different ages and genders are all at risk of suffering from anxiety disorders. Repeated and persistent stress [3], irritants, chronic diseases, and specific life stages (e.g., the perinatal period) [4] may lead to the development of anxiety disorders, which causes emotional and financial burdens to the patients. Common anxiolytic drugs, such as benzodiazepines, which are the most extensively studied at present, have side effects, such as sedation, memory disorders, tolerance, and withdrawal symptoms [5]. Therefore, it is necessary to develop alternative drug treatment strategies.

Zhi zhu xiang (ZZX) is the rhizome and root of *Valeriana jatamansi* Jones. It is a traditional Chinese medicine (TCM) that has the effect of calming nerves, according to the Chinese Pharmacopoeia. ZZX has been used to treat mental diseases worldwide [6], and has been confirmed to have a definite anxiolytic effect. The anxiolytic mechanism of ZZX may be related to the HPA axis [7,8], GABA level [9], cell apoptosis [10], etc. Based on our previous research, both 35% and 95% ethanol extracts of ZZX had significant anxiolytic effects, and the anxiolytic components of ZZX are iridoids, flavonoids, and phenolic acids. The anxiolytic compounds group (ACG) in a 35% ethanol extract of ZZX was screened by spectrum-effect correlation analysis [11], and the underlying mechanisms were explored using proteomics methods [12]. The main components in the 95% ethanol extract of ZZX are iridoids, which have been proven to have significant anti-anxiety effects, such as valtrate [7], which is unique to valerian plants. In our previous work, the spectrum-effect correlation method was used to identify the composition of ACG in 95% ethanol extracts of ZZX [11], including 11-ethoxyviburtinal, baldrinal, acevaltrate, and valtrate. However, the optimal dose and mechanism of action of ACG have not been evaluated so far.

Network pharmacology is an effective method that integrates chemical informatics, bioinformatics, network biology, and traditional pharmacology to reveal the bioactive components and complex mechanisms of TCM. It allows us to study the role of multiple compounds in biological networks and to explore multi-target mechanisms of TCM [13,14]. Metabolomics technology can be used to determine the potential mechanism of human diseases and the role of drugs in disease treatment by detecting and analyzing the changes in small-molecule metabolites and their correlation in the body [15,16]. The combination of these two technologies can well identify and connect the ACG, molecular targets, and metabolic effects, which is also suitable for elucidating the mechanism of multi-target drugs.

Therefore, in this study, the method of uniform design combined with mathematical modeling was used to optimize the proportion of ACG in a 95% ethanol extract of ZZX to achieve the best anxiolytic effect in vivo. The anxiolytic action pathway and core targets of the four compounds were predicted by network pharmacology, which were verified by molecular docking. Differential metabolites were screened in the rat hippocampus before and after the administration of ACG using metabolomics. Finally, integrated analysis of network pharmacology and metabolomics data was conducted to explore the mechanism of ACG in order to provide reliable ideas for the development of anxiolytic drugs.

## 2. Materials and Methods

### 2.1. Optimization of Anxiolytic Dose Ratio and Efficacy Evaluation

#### 2.1.1. Chemicals and Materials

ACG in a 95% ethanol extract of ZZX, including acevaltrate, valtrate, and baldrinal, was purchased from Sichuan Weikeqi Biological Technology Co., Ltd. (Weikeqi, Chengdu, China, HPLC, content ≥ 98%). 11-ethoxyviburtinal was extracted in the laboratory (HPLC, content ≥ 98%). Compounds were dissolved with 1% Tween-80 before use. ZZX was purchased from Jiudingshan, Sichuan, China (batch No. 2018S10), and was identified as the dried rhizome and root of *Valeriana jatamansi* Jones by Prof. Jinli Shi of Beijing University of Traditional Chinese Medicine—the voucher specimen (S180930) was deposited in the Herbarium of Beijing University of Traditional Chinese Medicine. The 95% ethanol extract of ZZX was used for the verification test, and the preparation process was as follows: 150 g of ZZX coarse powder was soaked in 1500 mL of 95% ethanol for 30 min and extracted at 40 °C using an ultrasonic system (3 × 1 h, power 500 W). The extracted solution was dried using a rotatory evaporator below 35 °C, and freeze-dried to a constant weight (17.72 g). The extract was dissolved with 1% Tween-80 before use.

#### 2.1.2. Animals

Specific-pathogen-free (SPF)-grade male Sprague Dawley (SD) rats, weighing 150–170 g (Beijing Charles River Laboratory Animal Technology Co., Ltd., Beijing, China), license No.: SCXK (Jing) 2016-0006), were individually housed in a standardized feeding environment with a 12 h light/dark cycle (light on 7:00–19:00) at 22 ± 2 °C and 60 ± 5% relative humidity. Before the experiment, the rats were provided standard feed and clean drinking water ad libitum. the experimental procedures were approved by the Animal Care and Use Committee of the Institute of Psychology of the Chinese Academy of Sciences (No. 20170327), and in compliance with the Provision and General Recommendation of Chinese Experimental Animals Administration Legislation.

#### 2.1.3. Establishment of the Rat EBS Model

The empty bottle stress (EBS) experiment was conducted for 21 days to establish an anxiety model [17]. EBS can simulate the psychological state in the process of anxiety disorder, which has the advantages of stability, reliability, easy repetition, and an obvious anxiety state. Except for the control group, the animals were given regular drinking training for 7 days. They were provided water for 10 min at 8:00–8:10 and 20:00–20:10 every day, and the water bottles were then removed. Starting at the 8th day, uncertain empty bottle stimulation was given within the two time periods above and maintained once/day or twice/day for 14 days, as shown in Figure 1A.

#### 2.1.4. Uniform Design Experiment

##### Animal Grouping and Drug Administration

In our previous study, the ACG in 95% ethanol extract of ZZX was screened, resulting in four components: 11-ethoxyviburtinal, baldrinal, acevaltrate, and valtrate. To calculate the best ratio of these four compounds, a uniform design was used for grouping. The administration grouping and dose proportion used in this experiment was designed according to the uniform design table U*8(8^5^) and the corresponding use table (Appendix A). Taking the content of the four compounds as the factor, the dose range was determined according to the content range of the four compounds in samples from different producing areas determined by the research group in the early stage. The design of the dose in the eight uniform designed groups (UDG) is shown in Table 1.

The rats were randomly divided into 11 groups: control group, model group, diazepam (DZP) group, and UDG (UDG1–UDG8) groups. After 7 days of water drinking training and 7 days of empty bottle stress, the control group, model group, DZP group, and UDG groups were administered drugs by gavage for 7 days (1 mL/100 g/day) from the 15th day. The control group and model group were given 1% Tween-80 solution, rats in the DZP group were given diazepam (1 mg/kg/day), and the UDG groups were given a mixture of four ingredients according to Table 1. During the administration period, uncertain empty bottle stimulation was given continuously. On the 22nd day, one hour after the last administration, the experimental rats in each group were selected in parallel for the behavioral test.

##### Elevated Plus-Maze Test (EPM)

The EPM has been widely used to evaluate potential therapeutic drugs for anxiolytic effects in rodents [18], and, in this study, the EPM was used to evaluate the anxiolytic effect of ACG. Before starting the EPM, each rat was allowed to explore freely for 5 min in a plastic box (45 cm × 30 cm × 15 cm) and then immediately placed at the central platform of the device (facing one of the open arms). An infrared video tracking system was used to automatically record the activities of rats within 5 min. After each test, the excreta of the rat was removed; then, the elevated maze was wiped with an alcohol cotton cloth, and dried with a dry cloth to remove the smell of rats. Data such as the duration and entries into the open arms and closed arms were collected. The percentage of entries into open arms (OE%) and the percentage of duration in the open arms (OT%) were used to evaluate the anxiolytic effect of UDGs.

##### Open Field Test (OFT)

The OFT has a solid theoretical rationale for detecting anxiety and the effectiveness of pharmacological agents [19]. Before the OFT, each rat was allowed to explore freely for 5 min in a plastic box, and then immediately placed at the edge area of the open field. The bottom of the open field was divided into the center region and peripheral region. An infrared video tracking system was used to automatically record the activities of rats within 5 min. After each test, the open field chamber was cleaned to remove the rat odor. The entries in the central area and time spent there were used to evaluate the anxiolytic effect of UDGs.

##### Sample Collection and Detection

HPA axis abnormalities and neurotransmitter disorders are two important mechanisms of anxiety and can also be used as biochemical indicators to determine anxiety [12]. Studies have shown that stress can reduce the level of N-arachidonoylethanolamide (AEA) in brain tissue, leading to the hyperfunction of the hypothalamic–pituitary–adrenal (HPA) axis and anxiety [20]. It has been shown that there is a correlation between the brain-derived neurotrophic factor (BDNF) levels and anxiety-related personality traits [21], and these indicators serve as components of a comprehensive pharmacodynamic indicator together. After the behavioral test, the rats were immediately anesthetized with 20% urethane, and blood was obtained through the abdominal aorta to obtain serum, and stored at −80 °C. After the rats were sacrificed, brain samples were rapidly isolated in a low-temperature environment and stored in an ultra-low-temperature freezer (−80 °C) for future analysis. The content of HPA axis-related hormones, corticotropin-releasing hormone (CRH), adrenocor ticotropic hormone (ACTH), and corticosterone (CORT) in the serum and the contents of dopamine (DA), 5-hydroxytryptamine (5-HT), norepinephrine (NE), γ-aminobutyric acid (GABA), AEA, and BDNF in the brains were measured using enzyme-linked immunosorbent assay (ELISA) kits (Shanghai Jianglai industrial Limited Co., Ltd.) according to the manufacturer’s instructions.

##### Determination of Optimal Proportion of ACGs

The optimal proportion was determined as described previously [12]. The contents of 11-ethoxyviburtinal, baldrinal, acevaltrate, and valtrate were taken as independent variables (X_1_–X_4_), and the comprehensive pharmacodynamics, including the index of EPM, OFT, and contents of CRH, ACTH, CORT, DA, 5-HT, NE, GABA, AEA, and BDNF, were calculated by entropy weighting as the dependent variable (Y) [12]. Minitab 19 statistical software was used for stepwise regression analysis and partial-least-squares regression (PLSR) analysis. Finally, the maximum values of the regression equation and PLSR equation were solved using the 1stOpt 8.0 software to obtain the optimal solution (OS), and the optimal proportion of the four pharmacodynamic components in the anxiolytic compound group of 95% ethanol extract of ZZX was obtained.

##### Verification Test

To verify the anxiolytic effect of ZZX-OS and to compare the accuracy of the calculation, rats were randomly divided into a control group (1% tween-80, 1 mL/100 g), model group (1% tween-80, 1 mL/100 g), DZP group (diazepam, 1 mg/kg/day), ZXX-OS groups, and 95% ethanol extract of ZZX group (1 g/kg/day, equivalent to the crude herb). The EBS-induced rat anxiety model combined with EPM and OFT was used for investigating the pharmacodynamics. The anxiolytic effects of OS and 95% ethanol extract of ZZX were compared.

##### Statistical Analysis of the Pharmacodynamic Experiment

SPSS 17.0 software was used to analyze the data. The results are expressed as the means ± SEM. One-way ANOVA with Dunnett’s multiple comparisons test was used to determine differences between groups. *^(#)^
*p* < 0.05 and **^(##)^
*p* < 0.01 were considered statistically significant.

### 2.2. Network Pharmacology Analysis

#### 2.2.1. Acquisition of Information and Targets of ACG Compounds

The relevant information of four compounds, including the molecular weight, SMILE numbers, two-dimensional (2D) structure, and three-dimensional (3D) structure, was obtained using the PubChem (https://pubchem.ncbi.nlm.nih.gov/, 16 September 2019) and CASC Databases (http://www.organchem.csdb.cn/scdb, 16 September 2019). The 2D and 3D structures were stored in the SDF format. We uploaded the SDF format files of acevaltrate, valtrate, baldrinal, and 11-ethoxyviburtinal to the Pharmmapper Database (http://www.lilab-ecust.cn/pharmmapper/, 16 September 2019) to predict the candidate target of each compound. The target results were also predicted by importing the SMILE number into the Swiss Target Prediction database (http://www.swisstargetprediction.ch/, 12 October 2019), followed by integrating the data to remove duplicates.

#### 2.2.2. Screening of Anxiety-Associated Targets

Anxiety disorder related-targets were collected from multiple databases, including PharmGKB (https://www.pharmgkb.org/, 11 October 2019), TTD (http://database.idrb.cqu. edu.cn/TTD, 11 October 2019), CTD (http://ctdbase.org/, 11 October 2019), Genecards (http: //www.genecards.org/, 11 October 2019), and Drugbank (http://www.drugbank.ca/, 11 October 2019). The keywords “anxiety” and “anxiety disorder” were used to search the reported genes, limiting the species to “Homo sapiens”, and integrated the target information related to anxiety. Duplicate genes and false-positive genes were removed.

#### 2.2.3. Construction of the PPI Network and Enrichment of Biological Function and Pathway

The official names of target genes were obtained from the UniProt database (http://www.uniprot.org/, 11 October, 2019), and Venny 2.1.0 (https://bioinfogp.cnb.csic.es/ tools/venny/index.html, 11 October 2019) was used to draw a Venn map of the target genes for the four compounds and anxiety-related target genes, resulting in common target genes. The protein–protein interaction (PPI) relationship between the common target genes was obtained using the STRING (http://STRING-db.org/, 12 October 2019) and GPS-Prot Databases (http://gpsprot.org/, 12 October 2019). The data were imported into Cytoscape 3.7.2 software to construct the PPI network of “anxiolytic ingredient groups–anxiety disease target gene”. Then, gene ontology (GO) and Kyoto Encyclopedia of Genes and Genomes (KEGG) pathway enrichment analyses were performed using the DAVID (https://david.ncifcrf.gov/, 14 January 2020) database, and the result was deemed significant at *p* < 0.05.

#### 2.2.4. Molecular Docking

To prove the accuracy of the network pharmacological prediction results, four compounds were used to dock with key target proteins. We used Open Babel 2.3.2 software to convert the SDF files into PDB files, and the receptor protein was retrieved from the Protein Data Bank database (http://www.rcsb.org/pdb, 15 March 2020). PyMOL 2.3.4 software was used to remove water and ligands, and MGLTools (http://mgltools.scripps.edu/, 15 March 2020) was used to perform hydrogenation and charge calculations on the screened receptor protein. AutoDock Vina was used to dock the receptor protein and ligand small molecules, take the best-scored conformation, and draw the docking with PyMOL 2.3.4 software.

### 2.3. Metabolomics Analysis

#### 2.3.1. Sample Preparation

The rats from the control group, EBS group, and ZZX-OS group were decapitated immediately after the behavioral test, and the cerebral cortex and hippocampus were quickly dissected on ice and stored at −80 °C. Before detection, the frozen samples were mixed with 25 mg of pre-chilled zirconium oxide beads and 10 μL of internal standard. An aliquot of 50% pre-chilled methanol (50 μL) was then added. Automatic homogenization was conducted, followed by centrifuging at 4 °C (14,000× *g* r/min) for 20 min. The supernatant was transferred into the autosampler vial. Then, 175 μL of pre-chilled methanol/chloroform (v/v = 3/1) solution was used for secondary extraction and centrifugation (4 °C, 14,000× *g* r/min). Then, 200 μL of the supernatant was transferred to a vial. The remaining supernatant was collected for quality control samples. All samples were evaporated using a vacuum concentrator to remove chloroform and further lyophilized using a freeze-dryer. Fifty microliters of methoxyamine was added to the dried sample, derivatized at 30 °C for 2 h, and then 50 μL of MSTFA (1% TMCS) containing FAMEs was added as retention indices at 37.5 °C for another 1 h.

#### 2.3.2. GC-MS Detection Conditions

The TOF/MS system (Pegasus HT, Leco Corp., St. Joseph, MO, USA) with an Agilent 7890B gas chromatography Rxi-5sil MS capillary column (30 mm × 250 μm, 0.25 μm, Restek Corporation, Bellefonte, PA, USA) was used, with helium as carrier gas and a constant flow rate of 1.0 mL/min. The temperature of the sample injection and transfer interface was set to 270 °C, and the source temperature was 220 °C. Detection in the full scan mode (m/z 50–500) was conducted with electron impact ionization (70 eV).

#### 2.3.3. Screening of Differential Metabolites and Analysis of In Vivo Metabolic Pathway

By comparing the retention index and mass spectrum data obtained by XploreMET software (Metabo-Profile, Shanghai, China) with the data generated by the reference standard of the known structure in the JiaLib metabolite database, the metabolites in the rat hippocampus were identified. Two statistical analysis methods were used: (1) multivariate statistical analysis, including principal component analysis (PCA) and orthogonal partial least-squares discriminant analysis (OPLS-DA), and (2) univariate statistical analysis, including Student’s *t*-test (*t*-test), the Mann–Whitney–Wilcoxon test (U-test), ANOVA, and the Kruskal–Wallis test. *p* < 0.05 and *p* < 0.01 were considered statistically significant. We integrated the results of multivariate and univariate statistical analyses to identify differential metabolites, and then inputted the potential biomarkers into Metaboanalyst to construct the metabolic pathways.

#### 2.3.4. Integration of Network Pharmacology and Metabolomics

The names of different metabolites in the rat hippocampus under the action of ACG in 95% ethanol extract of ZZX were imported into the KEGG database for ID conversion, and the ID information was input into Metscape (plug-in of Cytoscape software 3.7.2) to obtain the link proteins of differential metabolites, which were used as the action targets of differential metabolites. The target information obtained from metabolomics and network pharmacology was then imported into the STRING database and GPS Prot database for protein–protein interaction (PPI) analysis to visualize the PPI network, and the relevant targets were obtained to reveal the mechanism of ACG in the treatment of anxiety disorder.

## 3. Results

### 3.1. Elevated Plus-Maze Test (EPM)

The results of the EPM test are shown in Figure 1B, C and Appendix A. Compared to the control group, the OE% and OT% of rats in the EBS model group were significantly lower (*p* < 0.01). The OE% and OT% in the DZP group increased significantly compared with those of the EBS rats (*p* < 0.01). Except for the UDG-3, UDG-7, and UDG-8 groups, the UDG treatments significantly increased the OE% and OT% of rats (*p* < 0.01 or *p* < 0.05). In UDG-3, UDG-7, and UDG-8, the OT% value increased higher than that of the EBS group and followed the same trend as that of the other UDG groups. The EPM test in rats showed that the ACG in 95% ethanol extract of ZZX had a certain anxiolytic effect.

### 3.2. Open Field Test (OFT)

Compared to the control group, the number of central entries (CE) and time spent in the central areas (CT) by the EBS group were reduced (*p* < 0.05). Compared to the EBS group, DZP improved CE and CT (*p* < 0.05). The values of CT and CE for the UDG-1, UDG-4, UDG-7, and UDG-8 groups were significantly increased compared with that of the EBS group, (*p* < 0.01 or *p* < 0.05). The remaining UDG groups had higher scores. The results showed that the ACG in 95% ethanol extract of ZZX could alleviate anxiety-like behavior (Figure 1D,E and Appendix A).

### 3.3. Biochemical Indicators in Serum and Brain Samples

The results in Table 2 and Table 3 showed that the levels of CRH, ACTH, CORT, DA, 5-HT, and NE in EBS rats were significantly higher than those in the control rats (*p* < 0.01), while the levels of GABA, AEA, and BDNF were significantly decreased (*p* < 0.01). The pharmacodynamic indexes of the UDG groups (UDG-1–8) were statistically different from those of the EBS group (*p* < 0.05 or *p* < 0.01); their trends were similar to those of the DZP group (*p* < 0.01), indicating that the ACG with different doses and proportions have different anxiolytic effects.

### 3.4. Determination of the Best Proportion of Anxiolytic Components of 95% Ethanol Extract of ZZX

The stepwise regression equation of dose (X) and comprehensive efficacy (Y) was Y = 0.571 − 1.085 × 10^−8^X_4_^2^ + 20.786X_1_X_2_ + 5.398 × 10^−7^X_3_X_4_ − 7.146 × 10^−8^X_3_^2^. (*F* = 8.648, *p* < 0.05, *r* = 0.995), and the optimal solution (OS_1_) was: X_1_: 0.021, X_2_: 0.015, X_3_: 0.655, X_4_: 8.845, Unit: mg/kg.

The partial-least-squares regression equation was Y = 1422.91 − 36.03X_1_ + 222.81X_2_ − 8.8X_3_ + 1.99X_4_ − 116.57X_1_^2^ + 226.90X_1_X_2_ − 22.48X_1_X_3_ + 43.36X_1_X_4_ + 491.81X_2_^2^ + 5X_2_X_3_ + 37.63X_2_X_4_ − 0.59X_3_^2^ − 0.6X_3_X_4_ + 0.05X_4_^2^ (*p* < 0.05, *r* = 0.863). The optimal solution (OS_2_) obtained by solving the equation was: X_1_: 0.489, X_2_: 0.500, X_3_: 0.655, X_4_: 8.845, Unit: mg/kg.

### 3.5. Verification Test

In the EPM test, the values of OE% were higher in the OS_1_ and OS_2_ administration groups than that in the EBS group (*p* < 0.05). The values of OT% in the ZZX extract and OS_2_ groups were also increased (*p* < 0.05). There was no significant difference between the two OS groups, DZP group, and ZZX group. In the open field test, the indexes in the ZZX, OS_1_, and OS_2_ groups were significantly higher than those in the EBS group (*p* < 0.01). Although there was no significant difference between the three groups, the CT and CE of rats in OS_1_ and OS_2_ were somewhat higher than those of the ZZX group, and the OS2 group performed the best (Figure 2 and Appendix A). The results of the EPM test and open field experiment showed that the two optimal solutions (OS_1_ and OS_2_) had anxiolytic effects, which may have had the same anxiolytic effect as the extract, or even better than that of the extract.

### 3.6. Construction and Topology Analysis of PPI Network of Compounds–Anxiety Disease Target Genes

Based on network pharmacology analysis, 2434 anxiety targets and 206 ACG targets were obtained, and the number of common targets of compounds and diseases was 152. A Venn diagram of ACG target genes and anxiety-related target genes was drawn (Figure 3A). The PPI results of the 152 common targets were obtained using the STRING and GPS-Prot databases. The results were imported into Cytoscape software to construct the visual PPI network of the “ACG–anxiety disease target gene”, including 152 nodes and 1282 edges. A network analyzer was used for topology analysis (Figure 3C). Finally, 17 core targets were screened, and the results are shown in Table 4 and Figure 3B.

### 3.7. GO and KEGG Analysis

The GO function of 152 target genes mainly focused on the following aspects (Figure 4A): (1) regulation of cell proliferation and apoptosis, (2) DNA/RNA transcription regulation, (3) protein decomposition, synthesis, and phosphorylation, (4) signal transduction and cascade activation of multiple pathways, etc. KEGG pathway analysis enriched a total of 84 pathways, sorted by *p*-value, and retained 20 signaling pathways related to anxiety disorders (Figure 4B), in which the −log10 (*p*-value) of the estrogen and prolactin signaling pathways were the highest, showing that these two pathways play a crucial role in the anti-anxiety process of ACG. The enrichment results of KEGG pathway analysis are extensive, involving a variety of biological pathways (e.g., the synthesis, activation, transportation, and secretion of various proteins, cell growth, and differentiation) and biological molecules (e.g., prolactin, estrogen, insulin, thyroid hormone, and steroid hormone), which may be related to anxiety disorder. A Target–Pathway network was constructed, as shown in Figure 4C.

### 3.8. Molecular Docking Verification

The core target proteins AKT1, EGFR, ESR1, MAPK8, MMP9, and SRC that belong to the prolactin and estrogen signaling pathways from the PPI analysis were selected for molecular docking with four compounds in ACG, and their docking score was between −6.1 kcal/mol and −8.3 kcal/mol (Figure 5, Appendix A and Table 5), indicating that each chemical component was closely bound to the receptor protein. However, there were some differences in the binding ability between each compound and their respective receptor proteins.

### 3.9. Screening of Potential Biomarkers

The metabolites were identified using the XploreMET software, combined with previous literature and the JiaLib database. A total of 73 compounds were identified, and the metabolite names and category are shown in Appendix A. Both multivariate analyses (PCA and OPLS-DA) and univariate statistical analyses (*t*-test or Mann–Whitney U test) were conducted to screen potential biomarkers.

PCA analysis was used to obtain the principal component score diagram (PC1 = 21.1%, PC2 = 16.7%). Figure 6A shows that the metabolic profiles between the EBS model group and the control group are different. This difference indicates that the endogenous metabolites in the hippocampus of rats are either upregulated or downregulated after EBS modeling.

OPLS-DA models for the comparison between the two groups were established (Figure 6B,C). R^2^Y = 0.973, Q^2^ = 0.665, indicating that this model has a reliable predictive ability and can protect against over-fitting. Then, the marker metabolites were screened using a volcano plot based on the results of OPLS-DA. When the variable influence on projection (VIP) was >1.0, it was considered to be significantly different. The values of VIP and the correlation coefficients are shown in the volcano plot (Figure 6E and Appendix A). As shown in Figure 6E, a total of 26 differential metabolites were obtained.

To supplement the results of multivariate analyses, differential metabolites between the two groups were also obtained by univariate analysis (*t*-test or U test) based on the screening conditions (*p* < 0.05). The results are shown in the volcano plot (Figure 6D). Compared with the control group, the metabolites highlighted in the upper right corner were increased, and those highlighted in the upper left corner were decreased in the EBS group. Fifteen differential metabolites were obtained; compared with the control group, the metabolites of AMP, dehydroascorbic acid, D-Arabitol, Arachidonic acid, cellobiose, pyruvic acid, and palmitoleic acid were increased in the model group, whereas pentadecanoic acid, L-glutamic acid, behenic acid, N-methylalanine, DHA, sulfate, gluconolactone, and L-lactic acid were reduced, and they were all included in the 26 metabolites of OPLS-DA analysis.

To further elucidate the mechanism of ACG of 95% ethanol extract in ZZX, the metabolic profiles of the hippocampus in the control group, model group, and ZZX-OS_2_ group were analyzed by univariate analysis (ANOVA or Kruskal–Wallis test, *p* < 0.05). The abundances of seven differential metabolites in the administration group were relatively similar to those in the control group, indicating that the ACG had a callback effect on the metabolic disorder of anxiety rats caused by EBS. These seven metabolites were included in the 15 differential metabolites screened above. As shown in Figure 7, the ACG could significantly recall seven metabolites, including arachidonic acid, behenic acid, cellobiose, dehydroascorbic acid, L-leucine, n-methylalanine, and pentadecanoic acid (Table 6).

### 3.10. Enrichment of Metabolic Pathways

Metabolic pathway analysis of differential metabolites can reflect the biochemical disturbance of anxiety in rats induced by EBS, and may provide information for elucidating the action mechanisms. The information of the above seven metabolites was analyzed and four metabolic pathways were enriched, including arachidonic acid (ARA) metabolism, unsaturated fatty acid biosynthesis, glycolysis or gluconeogenesis, and pyruvate metabolism. The ARA metabolic pathway was the most significant one (impact value > 0.1). Because n-methylalanine, pentadecanoic acid, and cellobiose in differential metabolism could not be extracted in Metscape analysis, four metabolites were introduced into the Metscape plug-in to obtain 73 related protein targets (Appendix A).

### 3.11. Analysis of the Integration Mechanism of Metabolomics and Network Pharmacology

To comprehensively explore the potential mechanism of the anxiolytic ingredient group in 95% ethanol extract of ZZX in vivo, we integrated 73 differential metabolites obtained from metabolomics and 152 action targets obtained from network pharmacology. These data were imported into STRING and GPS-Prot databases to obtain PPI analysis data, and then imported into the Cytoscape software to visualize the PPI network (Figure 8). It was found that the ACG in 95% ethanol extract of ZZX played an anti-anxiety role by regulating the prolactin signaling pathway, estrogen signaling pathway, and ARA metabolism pathway, as well as affecting important targets such as ALB, AKT1, PTGS2, CYP3A4, ESR1, CASP3, CYP2B6, EGFR, SRC, MMP9, IGF1, and MAPK8.

## 4. Discussion

### 4.1. Dose Optimization of Anxiolytic Compounds Group

Compared with the medicinal materials and extracts, the effective compounds group (ECG) of traditional Chinese medicine has the advantages of clear components, controllable quality, and stable curative effect. To reasonably utilize effective compounds, it is necessary to optimize the dose of ECG administration. The key to optimization lies in the selection of the optimization method. In terms of experimental design, the uniform design not only fully considers the uniformity of experimental points, but also makes each experimental point more representative, can be used to minimize the occurrence of experimental errors, and improve the experiment efficiency on the premise of reducing the number of experiments [22]. In terms of regression models, partial-least-squares regression (PLSR) was designed to solve the problem of numerous possibly correlated predictor variables with few samples, and to deal with data with multicollinearity more effectively [23]. The stepwise regression method can eliminate insignificant variables and retain the factors that have a significant influence on the dependent variables, and the optimal regression model is ultimately derived [24]. The two mathematical modeling methods linked the dose and pharmacodynamic indicators and complemented each other to ensure the high accuracy of the prediction results.

In this study, a uniform design combined with PLSR and stepwise regression was used to optimize the best proportion of four anxiolytic compounds in 95% ethanol extract of ZZX: 11-ethoxyviburtinal (0.489 mg/kg), baldrinal (0.500 mg/kg), acevaltrate (0.655 mg/kg), and valtrate (8.845 mg/kg). In recent years, the experimental design combined with mathematical modeling has been used to optimize the proportion of multi-drug combinations, and is an effective, multi-objective optimization method [25,26]. Our study provides a reference for solving such problems. We confirmed the anxiolytic effect of ACG using behavioral experiments and biochemical indicators. The results revealed that ACG could regulate HPA axis hormones and neurotransmitters in the brain to reduce anxiety-like behavior in EBS rats.

### 4.2. Mechanism Prediction and Verification by Network Pharmacology and Molecular Docking

Based on the significant anxiolytic effect of ACG in 95% ethanol extract of ZZX, the mechanisms underlying its effect were further explored. Network pharmacology could predict the possible pathways involved in the drug treatment effects and reveal the relationships between compound targets and disease-associated proteins [27], especially for multi-target drugs. In our study, 17 core common targets of ACG and anxiety were predicted, of which ESR1, SRC, AKT1, MAPK8, EGFR, and MMP9 belong to the prolactin and estrogen signaling pathways. Molecular docking was performed to verify the accuracy of prediction. The results showed that the four compounds in ACG had good binding ability with the target proteins. Estrogen receptor 1 (ESR1) is an estrogen receptor subtype, and genetic polymorphisms in ESR1 have been proven to be associated with anxiety in humans [28]. Sarcoma tyrosine kinase (SRC) could mediate the phosphorylation of the NMDAR complex [29], and NMDAR2B phosphorylation could trigger anxiety-like behavior by regulating amygdaloid CRF expression [30]. Thus, it is recognized as a modulator of neurotransmitter receptor function and behavior [31]. MAPK8 (mitogen-activated protein kinase 8), also known as JNK1 [32], can be activated by exposure to environmental stresses [33], and it regulates the proinflammatory cytokine levels in the central nervous system [34]. Therefore, we speculate that ACG can affect stress-induced anxiety behavior by regulating MAPK8. In a pressure-induced rat model, matrix metalloproteinase-9 (MMP9) had abnormal expression. Similarly, the single nucleotide polymorphisms (SNPs) in the human homologs of MMP9 were significantly associated with susceptibility to anxiety disorders [35]. Although AKT1 and EGFR were predicted to be involved in anti-anxiety processes in our study, the two targets have not been shown to be directly involved in the anti-anxiety process. However, they may indirectly affect anxiety-like behavior.

The prolactin and estrogen signaling pathways have been extensively studied in the field of anxiety disorders. It has been reported that a combination of prolactin and its receptor can activate the MAPK pathway to mediate the phosphorylation of ERK, affect the expression of CRH, and regulate the response of the HPA axis [36]. Prolactin can also regulate the synthesis and release of dopamine to participate in the anxiolytic effect [37]. The estrogen pathway has been shown to play an important role in anxiety disorder. For example, substantial evidence shows that low estrogen levels influence anxiety in women [38,39]. Animal studies have confirmed that the loss of estrogen and estrogen receptors can lead to anxiety-like behavior [40,41]. The estrogen system can influence anxiety through HPA axis mediation, and it can also regulate DA receptors, influence 5-HT, GABA, OT system [38], etc. Estrogen can activate the MAPK signaling pathway, PI3K-AKT signaling pathway, and adenylate cyclase (cAMP) by binding with estrogen receptors [42,43]. It is noteworthy that prolactin mediates the production and secretion of estrogen [44]. Thus, the two pathways and related targets could regulate the HPA axis and neurotransmitters to provide an anxiolytic effect. Considering the high prevalence of anxiety disorder in women [45], further analysis of the estrogen and prolactin pathways has practical significance in the treatment of anxiety disorder.

### 4.3. Mechanism Exploration by Metabolomics

Although network pharmacology has been considered as an effective method to complement the established pharmacological approaches [27], the possible mechanisms by which drugs work in vivo are complex and varied, and the results obtained by a single technique may not be comprehensive. Therefore, the metabonomics approach was used to explore the anti-anxiety mechanism of ACG. A total of 73 metabolites were identified by metabonomics, and 15 potential biomarkers were screened by univariate and multivariate statistical analyses. Among them, the values of seven different metabolites in the administration group were relatively similar to those in the control group. It was found that ARA metabolism was significantly affected. Clinical research and animal experiments have confirmed that ARA is associated with anxiety disorder. For instance, the plasma levels of ARA in anxiety patients changed significantly compared with those in the normal group [46]. Animal experiments demonstrated that voluntary running in mice could reduce anxiety-like behavior by increasing the accumulation of ARA in the cerebral cortex [47]. Moreover, the ARA metabolite is also the precursor of endocannabinoid (eCB) [48], and eCB affects anxiety-like behavior by activating the endocannabinoid system and regulating neurotransmitters, CRH in the HPA axis, and the dopaminergic reward system in the brain [49]. Thus, regulating endogenous ARA by drug administration or exercise may be a viable method to treat or prevent psychiatric illnesses.

### 4.4. Mechanism Exploration by Integrating Network Pharmacology and Metabolomics

To systematically and comprehensively elucidate the anxiolytic mechanism of the ACG in 95% ethanol extract of ZZX, the targets of network pharmacology and differential metabolites were integrated to construct a PPI network of common targets (Figure 8). These targets were ranked according to their correlation with anxiety disorders by Cytoscape. The results showed that ALB, AKT1, PTGS2, CYP3A4, ESR1, CASP3, CYP2B6, EGFR, SRC, MMP9, IGF1, and MAPK8 played important roles in the treatment of anxiety disorders by ACG. Additional targets have been shown to be associated with anxiety disorders, for example, psilocybin, a drug with therapeutic potential for anxiety, which reduces the expression of prostaglandin-endoperoxide synthase 2 (PTGS2) in the hippocampus [50]. However, it remains unknown whether PTGS2 is directly related to anxiety. Cytochrome P450 (CYP) 3A4 inhibitor significantly increases the anti-anxiety effect of Tandospirone by increasing the plasma concentration of Tandospirone [51]. Insulin-like growth factor 1 (IGF1) has been shown to be the only growth factor with anxiolytic and antidepressant properties in human clinical trials [52]. However, there is no evidence that ALB, CASP3, and CYP2B6 are directly related to anxiety. Therefore, their specific mode of action needs to be further studied. According to the above research, the Target-signaling-pathway-regulatory index–metabolite regulatory network of ZZX-ACG was constructed. As shown in Figure 9, ACG has an anti-anxiety effect by regulating the prolactin signaling pathway, estrogen signaling pathway, and ARA metabolism pathway through ALB, AKT1, PTGS2, CYP3A4, ESR1, CASP3, CYP2B6, EGFR, SRC, MMP9, IGF1, and MAPK8, thus directly or indirectly affecting the levels of neurotransmitters in the brain and peripheral HPA axis hormones. This study could systematically and clearly sort out the drug action targets from the complex physiological and pathological processes of the body and accurately grasp the regulation objects, providing a reasonable and effective method for the study of the mechanism of anxiety disorder.

## 5. Conclusions

The proportion of the anxiolytic compounds group in 95% ethanol extract of ZZX were determined to be 11-ethoxyviburtinal (0.489 mg/kg), baldrinal (0.500 mg/kg), acevaltrate (0.655 mg/kg), and valtrate (8.845 mg/kg). Our results showed that ACG had a significant anti-anxiety effect, which was superior to that of the extract of the raw materials. Through ALB, AKT1, PTGS2, CYP3A4, ESR1, CASP3, CYP2B6, EGFR, SRC, MMP9, IGF1, MAPK8, and other targets, ACG in the 95% ethanol extract of ZZX could regulate the prolactin signaling pathway, estrogen signaling pathway, and ARA metabolism pathway, thereby directly or indirectly affecting the brain neurotransmitter levels and HPA axis hormone levels to exert an anti-anxiety mechanism. This study provides a scientific basis for the development of ZZX as a safe and reliable anxiolytic drug.

## Figures and Tables

**Figure 1 brainsci-12-00589-f001:**
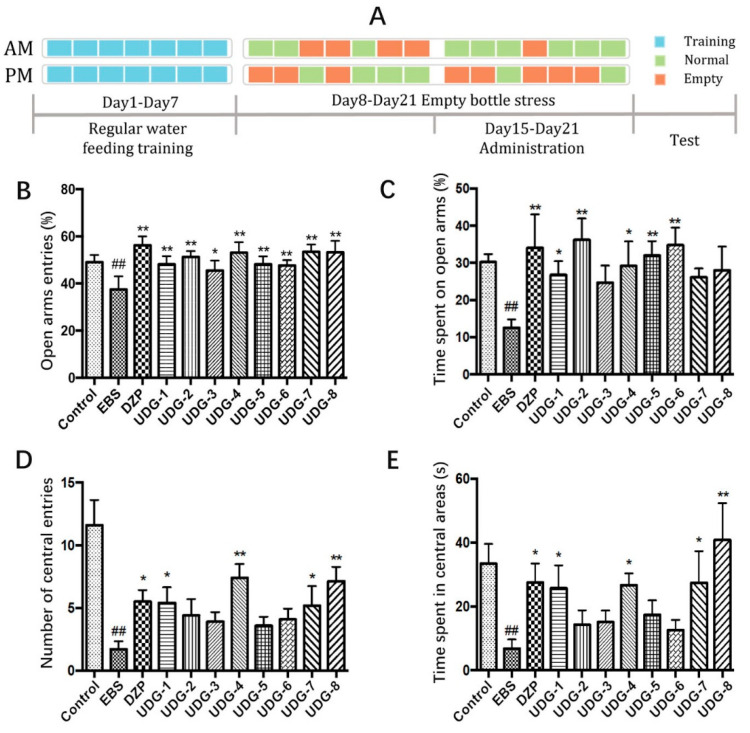
Anxiolytic effect of ZZX-UDGs on EBS rats. (**A**) Schematic diagram of the experimental process. (**B**) Effect of ZZX-UDGs on the open-arm entries (OE%) in the EPM test in EBS rats. (**C**) Effect of ZZX-UDGs on the time spent in the open arms (OT%). (**D**) Effect of ZZX-UDGs on the number of central entries (CE) in OFT in EBS rats. (**E**) Effect of ZZX-UDGs on the time spent in central areas (CT) in OFT. (*n* = 8–10, mean ± SEM. ^##^ *p* < 0.01 vs. Control; * *p* < 0.05, ** *p* < 0.01 vs. EBS).

**Figure 2 brainsci-12-00589-f002:**
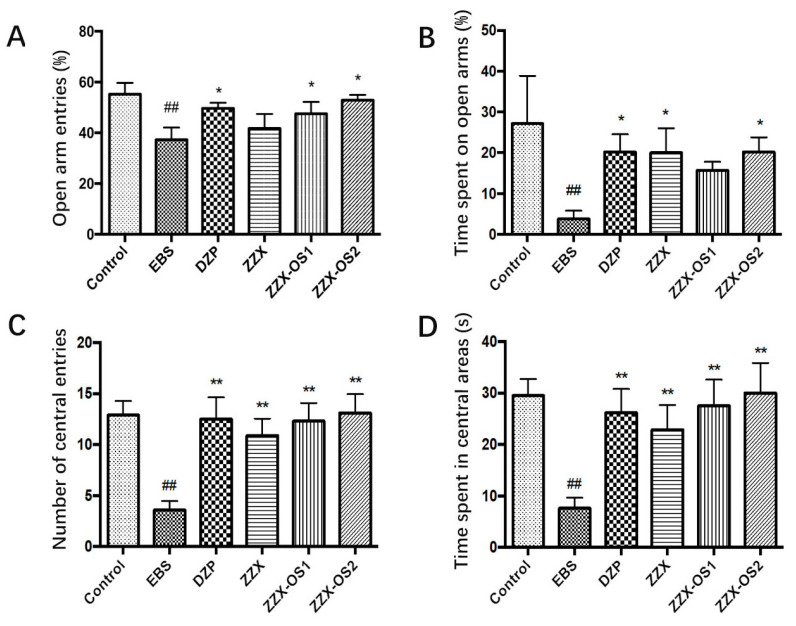
Comparison of the anxiolytic effect of the two optimal solutions (OSs) and ethanol extract of ZZX. (**A**) Open-arm entries (OE%). (**B**) Time spent in the open arms (OT%) (*n* = 6–8). (**C**) Number of central entries. (**D**) Time spent in central areas. (*n* = 10; mean ± SEM. ^##^ *p* < 0.01 vs. Control; * *p* < 0.05, ** *p* < 0.01 vs. EBS).

**Figure 3 brainsci-12-00589-f003:**
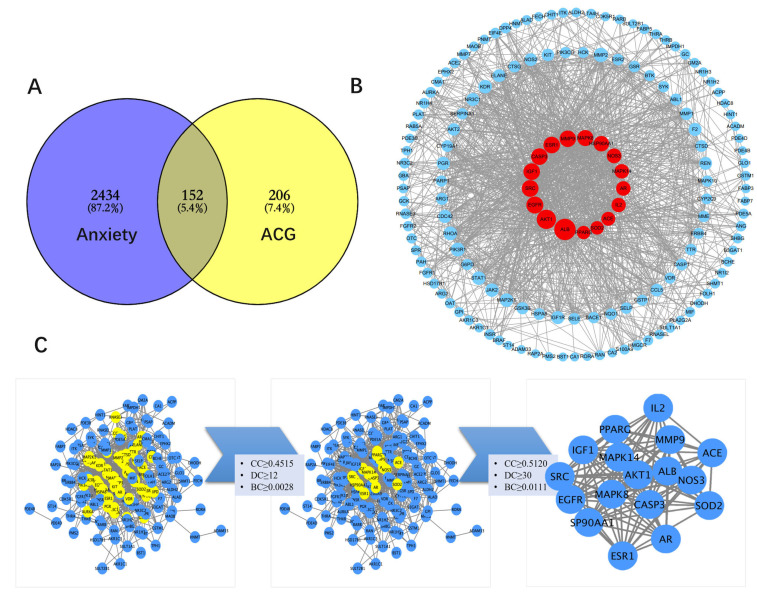
PPI network results of compounds–anxiety disease target genes. (**A**) Venn diagram of ACG target genes and anxiety-related target genes. (**B**) Screening of 17 core common targets. (**C**) Results of topology analysis of the PPI network. The first analysis results were CC (closeness centrality) ≥ 0.4515, DC (degree centrality) ≥ 12, and BC (betweenness centrality) ≥ 0.0028. A total of 54 nodes and 598 edges were reserved. The results of the second topology analysis indicated CC ≥ 0.5120, DC ≥ 30, and BC ≥ 0.0111. A total of 17 nodes and 119 edges were reserved.

**Figure 4 brainsci-12-00589-f004:**
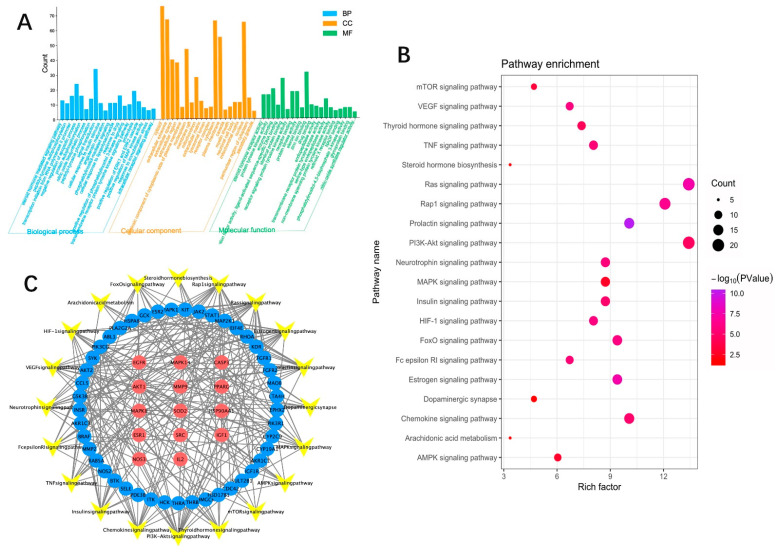
(**A**) GO function enrichment of ZZX-ACG–anxiety-related target. BP: Biological Process. CC: Cellular Component. MF: Molecular Function. (**B**) KEGG enrichment of ZZX-ACG–anxiety-related target. (**C**) Target–Pathway network of ZZX-ACG for anxiety.

**Figure 5 brainsci-12-00589-f005:**
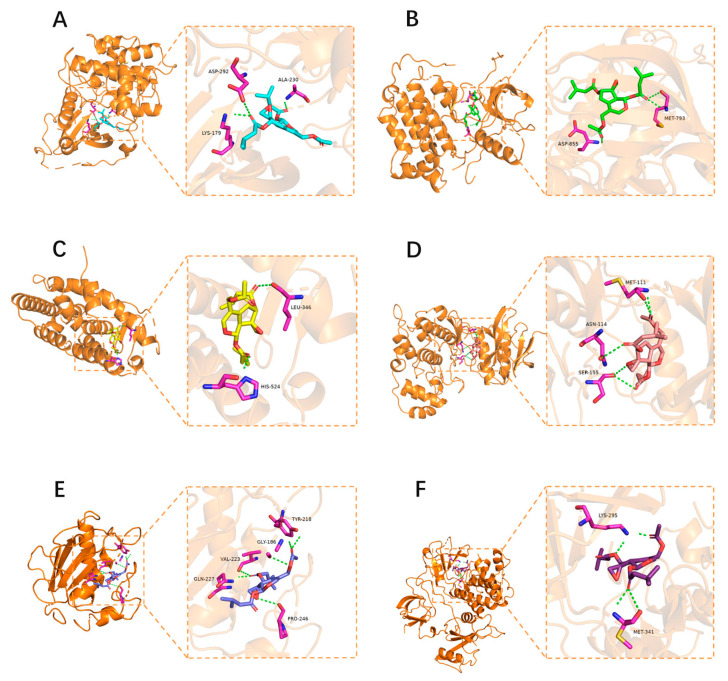
Molecular docking analysis of valtrate and 6 core target proteins. (**A**) AKT1, (**B**) EGFR, (**C**) ESR1, (**D**) MAPK8, (**E**) MMP9, and (**F**) SRC.

**Figure 6 brainsci-12-00589-f006:**
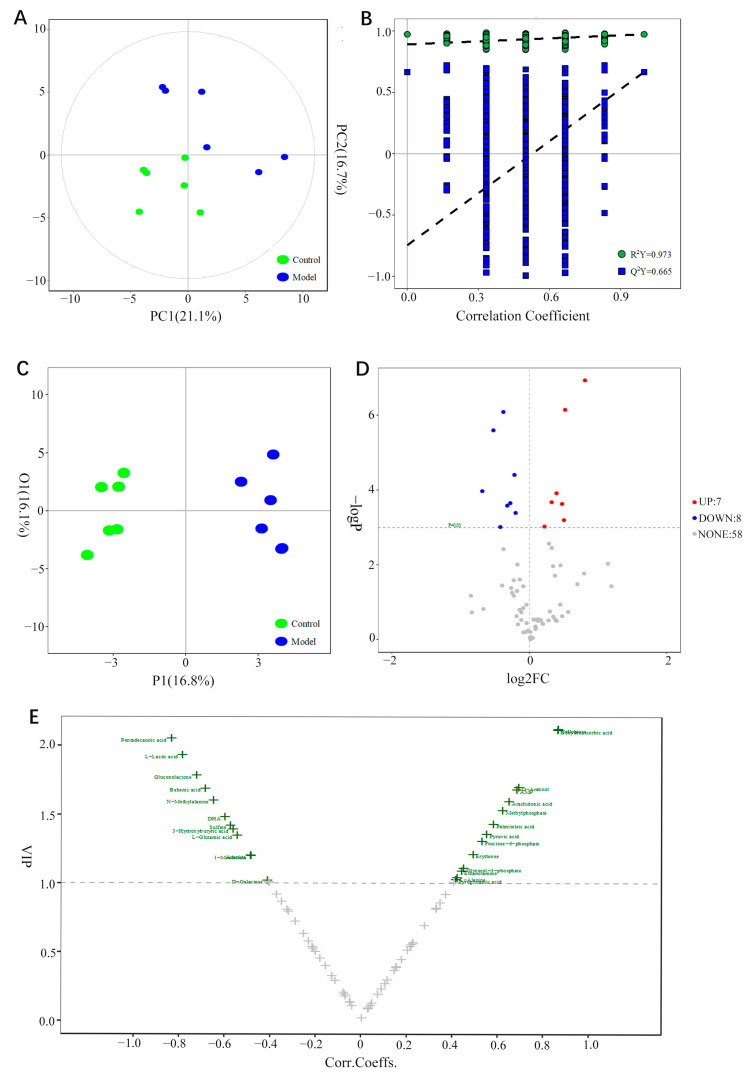
Potential biomarkers in rat hippocampus screened by univariate and multivariate statistical analyses. (**A**) PCA score plot of the control and model. (**B**) Results of permutation test. (**C**) OPLS-DA score plot of control and model. (**D**) Volcano plot of differential metabolites screened by univariate statistical analyses. (**E**) Volcano plot of differential metabolites screened by multivariate statistical analyses (OPLS-DA).

**Figure 7 brainsci-12-00589-f007:**
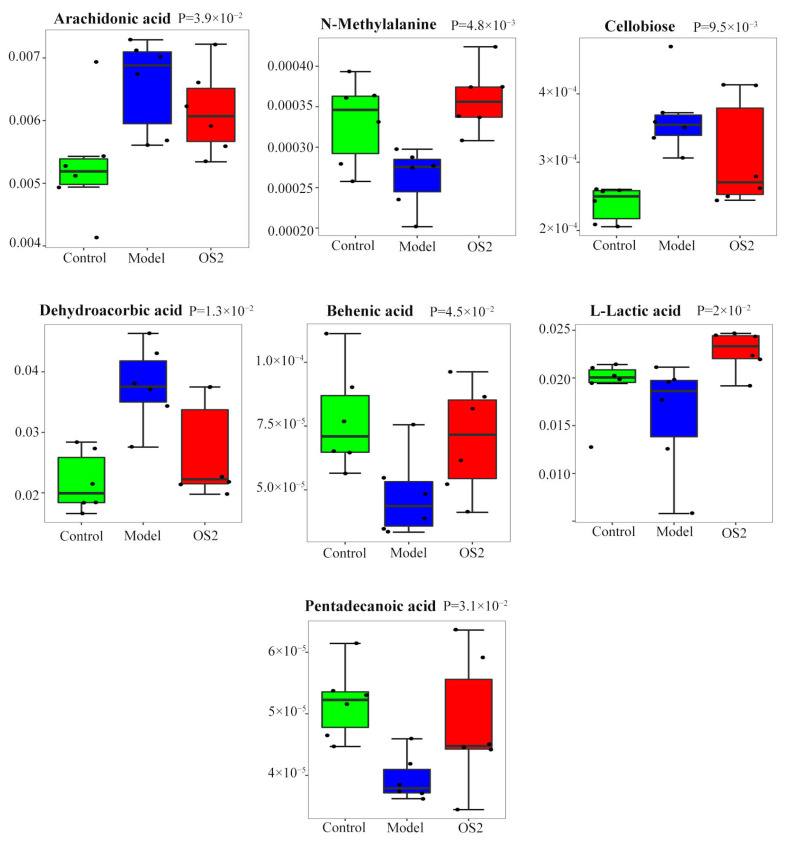
Differential metabolites in anxiety rats treated with ACG in 95% ethanol extract of ZZX.

**Figure 8 brainsci-12-00589-f008:**
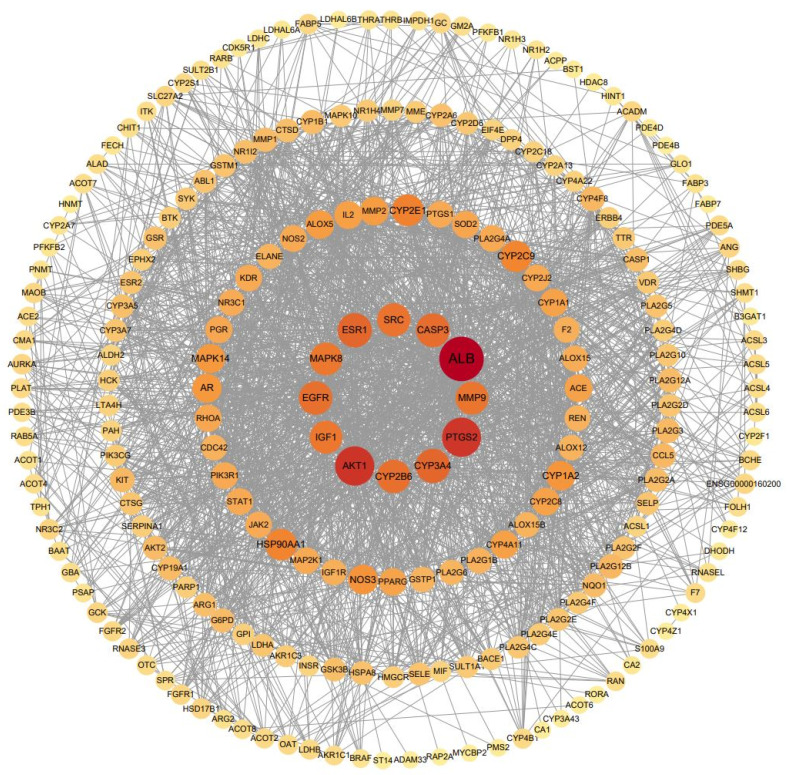
PPI network of the metabolomics and network pharmacology comprehensive targets.

**Figure 9 brainsci-12-00589-f009:**
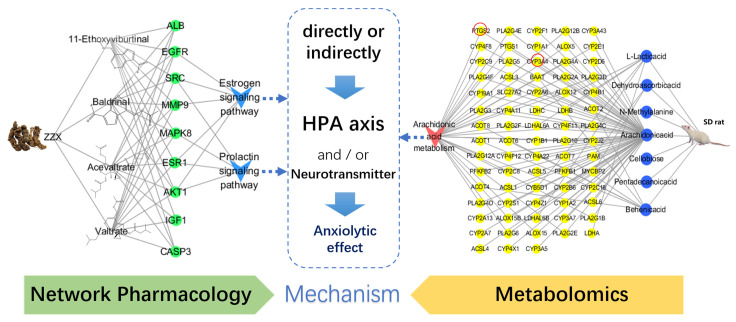
An illustration of the target-signaling-pathway-regulatory index–metabolite regulatory network.

**Table 1 brainsci-12-00589-t001:** Uniform designed group dosing scale (mg/kg/day).

Groups	11-Ethoxyviburtinal	Baldrinal	Acevaltrate	Valtrate
UDG-1	0.0210	0.0843	6.1004	8.8450
UDG-2	0.0879	0.2229	13.3610	7.6340
UDG-3	0.1547	0.3614	4.2853	6.4230
UDG-4	0.2216	0.5000	11.5459	5.2120
UDG-5	0.2884	0.0150	2.4701	4.0010
UDG-6	0.3553	0.1536	9.7307	2.7900
UDG-7	0.4221	0.2921	0.6550	1.5790
UDG-8	0.4890	0.4307	7.9156	0.3680

**Table 2 brainsci-12-00589-t002:** Uniform designed experiment—multiple pharmacodynamic indicators in serum (*n* = 8–10; mean ± SEM).

Groups	CRH (pg/mL)	ACTH (pg/mL)	CORT (ng/mL)
Control	154.40 ± 14.86	39.59 ± 2.53	6.98 ± 0.54
EBS	390.17 ± 9.21 ^##^	81.41 ± 2.69 ^##^	18.58 ± 0.64 ^##^
DZP	221.20 ± 10.18 **	48.57 ± 2.00 **	10.57 ± 0.56 **
UDG-1	337.93 ± 10.77 **	74.48 ± 2.68 *	16.73 ± 0.69 *
UDG-2	317.92 ± 12.64 **	66.23 ± 2.83 **	13.21 ± 0.56 **
UDG-3	272.24 ± 11.12 **	53.96 ± 2.24 **	12.60 ± 0.57 **
UDG-4	318.58 ± 13.93 **	68.93 ± 1.93 **	14.59 ± 0.71 **
UDG-5	219.45 ± 15.28 **	49.35 ± 1.94 **	9.03 ± 0.31 **
UDG-6	281.65 ± 12.29 **	55.17 ± 2.57 **	11.89 ± 0.45 **
UDG-7	227.29 ± 11.99 **	57.30 ± 1.84 **	10.68 ± 0.43 **
UDG-8	249.70 ± 9.46 **	51.46 ± 2.25 **	11.28 ± 0.38 **

^##^ *p* < 0.01 vs. control group, * *p* < 0.05, ** *p* < 0.01 vs. EBS group.

**Table 3 brainsci-12-00589-t003:** Uniform designed experiment—multiple pharmacodynamic indicators in brains (*n* = 8–10; mean ± SEM).

Groups	DA (pg/mL)	NE (ng/mL)	5-HT (ng/mL)	AEA (pg/mL)	GABA (μmol/L)	BDNF (pg/mL)
Control	2236.63 ± 511.43	41.57 ± 1.23	116.07 ± 8.00	858.43 ± 35.57	0.08 ± 0.00	6909.32 ± 271.64
EBS	9088.27 ± 687.71 ^##^	78.59 ± 2.11 ^##^	248.17 ± 11.28 ^##^	305.42 ± 17.56 ^##^	0.04 ± 0.00 ^##^	2340.67 ± 125.69 ^##^
DZP	3439.27 ± 353.16 **	46.33 ± 2.45 **	135.78 ± 4.69 **	759.61 ± 24.88 *	0.07 ± 0.00 **	5259.82 ± 319.53 **
UDG-1	7450.23 ± 613.73 *	66.51 ± 1.97 **	228.36 ± 5.59 *	376.09 ± 19.89 **	0.05 ± 0.00 *	3229.53 ± 221.48 **
UDG-2	5273.24 ± 560.59 **	50.10 ± 2.34 **	158.03 ± 5.76 **	505.42 ± 33.21 **	0.06 ± 0.00 **	3378.58 ± 218.25 **
UDG-3	6115.79 ± 713.41 **	53.56 ± 3.01 **	173.86 ± 8.18 **	650.88 ± 31.15 **	0.06 ± 0.00 **	4989.29 ± 272.07 **
UDG-4	7310.78 ± 417.66 *	62.17 ± 2.84 **	214.11 ± 5.96 **	669.44 ± 33.61 **	0.06 ± 0.00 **	4690.22 ± 284.78 **
UDG-5	5798.07 ± 34.47 **	54.96 ± 0.24 **	176.35 ± 0.39 **	592.23 ± 2.24 **	0.06 ± 0.00 **	4234.74 ± 19.98 **
UDG-6	5512.12 ± 576.85 **	56.78 ± 2.59 **	188.06 ± 6.09 **	594.20 ± 3.09 **	0.06 ± 0.00 **	3949.78 ± 263.03 **
UDG-7	6962.47 ± 420.10 **	62.98 ± 0.99 **	184.72 ± 4.59 **	596.73 ± 1.57 **	0.06 ± 0.00 **	4861.39 ± 265.60 **
UDG-8	5806.90 ± 44.41**	54.53 ± 0.23 **	176.19 ± 0.25 **	591.40 ± 2.70 **	0.06 ± 0.00 **	4177.36 ± 26.76 **

^##^ *p* < 0.01 vs. control group, * *p* < 0.05, ** *p* < 0.01 vs. EBS group.

**Table 4 brainsci-12-00589-t004:** Screening results of ZZX–ACG–anxiety disease PPI core targe.

No.	Gene Symbol	Closeness Centrality	Degree Centrality	Betweenness Centrality
1	ALB	0.68981481	89	0.18587974
2	AKT1	0.64224138	74	0.08939001
3	EGFR	0.58893281	58	0.04228520
4	SRC	0.57088123	56	0.05227749
5	IGF1	0.57976654	55	0.02219982
6	ESR1	0.58431373	55	0.05721287
7	CASP3	0.59126984	55	0.03857080
8	MMP9	0.58203125	54	0.03623026
9	MAPK8	0.57528958	51	0.01905757
10	HSP90AA1	0.56439394	47	0.03408826
11	NOS3	0.55805243	42	0.02089920
12	AR	0.54779412	40	0.02311410
13	MAPK14	0.54578755	40	0.01428905
14	IL2	0.53985507	39	0.02361781
15	ACE	0.53024911	36	0.01298736
16	SOD2	0.53214286	33	0.04665146
17	PPARG	0.52836879	32	0.02068498

**Table 5 brainsci-12-00589-t005:** Docking scores of four components with multiple receptor proteins.

Protein	PDB ID	Grid Size	Docking Score (kcal/mol)			
			Acevaltrate	Valtrate	Baldrinal	11-Ethoxyviburtinal
ESR1	6VJ1	42 × 44 × 44	−7.1	−7.8	−7.0	−6.4
SRC	2PTK	62 × 58 × 66	−8.3	−8.3	−7.1	−6.6
AKT1	4EKL	40 × 40 × 56	−8.1	−8.0	−7.0	−6.7
MAPK8	3VUD	58 × 58 × 62	−7.6	−7.1	−6.6	−6.4
EGFR	6S9C	46 × 48 × 66	−7.3	−8.2	−6.7	−6.1
MMP9	6ESM	42 × 34 × 40	−8.2	−8.2	−8.1	−7.4

**Table 6 brainsci-12-00589-t006:** Regulation of metabolites by ZZX-ACG in the hippocampus of anxiety rats.

No.	Metabolites	*p*-Value	Trend
1	L-Lactic acid	0.019994847	↑
2	N-Methylalanine	0.004791963	↑
3	Behenic acid	0.045431849	↑
4	Pentadecanoic acid	0.030987525	↑
5	Cellobiose	0.009514191	↓
6	Arachidonic acid	0.039333617	↓
7	Dehydroascorbic acid	0.013123729	↓

↑: increased vs. EBS group, ↓: decreased vs. EBS group.

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
