# Peer review of "Dose Optimization of Anxiolytic Compounds Group in Valeriana jatamansi Jones and Mechanism Exploration by Integrating Network Pharmacology and Metabolomics Analysis"

_brainsci, 2022, doi:10.3390/brainsci12050589_

Round 1
Reviewer 1 Report
The manuscript is relevant and presents important findings concerning the dose ratio of each component of anxiolytic compounds group (ACG) in 95% ethanol extract of Valeriana jatamansi Jones (Zhi zhu xiang, ZZX) was optimized by uniform design experiment and mathematical modeling.
The title of the work is clear and reflects the findings, as well as the objectives.
However, some extra information’s and corrections are needed:
Minor:
- Some words need to be spelled out the first time they are used in the body of the text, for example: ZZX (line 47); SPF male SD rats (line 98); CRH, ACTH, and CORT in serum and the contents of DA, 5-HT, NE, GABA, AEA, and BDNF (line 178 and 179).
- For a better presentation and understanding of the results, I suggest dividing Table 2 into: Indicators in Serum (2A) e Indicators in Brain (2B).
- Review and standardize the statistical significance value, see example line 337 and 341 #p < 0.05, ##p < 0.01 vs control group, *p < 0.05, *p < 0.01 vs EBS group. Sometimes the * is compared to the control, sometimes to the EBS group. This must be reviewed for all results.
- For better understanding and clarity of the data presented, it is necessary to insert the name of the statistical test used in each protocol and in the results.
Reviewer 2 Report
-how was the empty bottle stimulation randomized for uncertain days?
-for the pharmacodynamic indices (tables 2 and 3), did you control for multiple comparisons to prevent a false positive?
-can you provide the VIP in supplementary information of the OPLS-DA plot? Thank you for providing the VIP for the volcano plot.
-is the screening cutoff of a log2 > 0 stringent enough for your metabolites? Generally something greater is used reflecting a 2-fold increase or decrease.
-I am not clear on how you chose significant metabolites for entry into your other models (metabolic pathway analysis and integration of metabolomics and network pharmacology)? Is this from your univariate analyses which are based on your volcano correlational analysis? Also, since volanco plots just show correlation analysis, wouldn't this be considered a univariate analysis? Were there any overlap in metabolites identified as important for the PCA and OPLSD-A plots and also in the volcano plot? Just more clarification on how you go from multi-variate to univariate to your final set of metabolites of significance that you use in your subsequent analyses would be useful.
